# Pandemic Portraits—An Intersectional Analysis of the Experiences of People with Disabilities and Caregivers during COVID-19 in Bangladesh and Liberia

**Shahreen Chowdhury** [1,*] **, Salma Akter Urme** [2] **, Boakai A. Nyehn, Jr.** [3] **, Heylove R. Mark, Sr.** [3] **,**
**Md. Tanvir Hassan** [2] **, Sabina F. Rashid** [2] **, Naomi B. Harris** [3] **and Laura Dean** [1]

1   Liverpool School of Tropical Medicine, Liverpool L3 5QA, UK
2   BRAC James P Grant School of Public Health, BRAC University, Dhaka 1213, Bangladesh
3   National Union of Disabled Liberia, 19th Street Sinkor, Coleman Avenue, Monrovia 1000, Liberia
*   Correspondence: shahreen.chowdhury@lstmed.ac.uk

**Abstract:** COVID-19 significantly affected people with disabilities, with many facing additional barriers in access to services and increased risks of poor health and social outcomes. Focusing on the impact of COVID-19 in the Global South, this study took place in Bangladesh and Liberia, where 14% and 16% of the population are thought to live with disabilities. However, there is minimal research on the needs and experiences of this population group and how these are shaped by intersecting axes of inequity. Furthermore, disabled people are often excluded from being actively involved in research. To address these evidence gaps, we used the creative participatory method of photovoice remotely to document experiences of COVID-19 through the lens of people with physical and psychosocial disabilities and their caregivers as co-researchers. The findings present themes relating to inaccessibility, social connection, hopes and fears. The nexus between disability and poverty was exacerbated for many in both settings, while psychosocial impacts of COVID-19 included increased stigmatisation and isolation. However, themes of faith, support and adaptability were also highlighted in stories of community care, nature and healing. Photovoice, through imagery and storytelling, was a powerful tool in prioritising the voices of disabled people, adding to an evidence base to inform inclusive pandemic responses.

**Keywords:** COVID-19; disability; inclusion; intersectionality; psychosocial; Bangladesh; Liberia; pandemic; photovoice

## 1. Introduction

While COVID-19 presents challenges worldwide, the pandemic has disproportionately affected vulnerable groups, including people with disabilities (Hankivsky and Kapilashrami 2020). During the pandemic, as in other times of crisis, many people with disabilities have faced additional barriers in access to services, increased isolation and increased risks of poor health and social outcomes (Tuffrey-Wijne et al. 2014). Further, disabled people are particularly vulnerable to the impacts of COVID-19 due to intersecting biological and social risk factors, including: higher prevalence of co-morbidities and chronic conditions and the necessity of contact with caregivers, relatives and support staff (Courtenay and Perera 2020). For many people with disabilities, the nexus between disability and poverty has also been exacerbated, with many already living close to the margins of poverty due to reduced incomes or unemployment (Banks et al. 2017; Dean et al. 2018). Collectively, the physical and social impacts of COVID-19 exacerbate the risk of stigmatisation and neglect of groups who are often already socially isolated, cumulating in negative impacts on the mental wellbeing for these populations. For example, people with cognitive disabilities in the USA have been reported to experience heightened stress due to changes in routines and isolation as a result of social-distancing measures (Torres 2020). Such stresses are further

heightened when existing services for psychosocial support are also disrupted, which is common during crises; this, coupled with further stigma and neglect, can expose people with disabilities to experience abuse and violence (Ryan 2020).

The majority of the world's disabled population, estimated around 80%, live in low- and middle-income countries (LMICs). However, much of disability research focuses on the Global North, with a gap in evidence across LMICs (Grech 2016). In LMICs, people with disabilities are often also living in conditions of poverty. This further adds to the social determinants of poor health outcomes, particularly in the Global South, as many rely on daily income and live in informal settlements, with less access to care and, often, the absence of services (Emerson 2011); this is often worsened during health system shocks, such as pandemics. Therefore, this study aimed to capture and understand the lived, individual experiences of people with disabilities and their caregivers during COVID-19 in real time across two different LMIC settings in Bangladesh and Liberia. In Bangladesh and Liberia, 14% and 16% of the population, respectively, are thought to live with disabilities (ILO 2020; LISGIS 2017). However, there is minimal academic research which documents the needs and experiences of this population group, and an even greater sparsity of studies that consider how these experiences are shaped by intersecting axes of inequity, such as age, gender, socio-economic status or type of impairment. Furthermore, people with disabilities are often excluded from being actively involved in research, particularly those who live at the intersection of marginalised identities, for example, vulnerable groups such as disabled women who may be displaced and live in remote, resource poor settings (Banks et al. 2017; Durrell 2016). Thus, understanding the impact of COVID-19 on people with disabilities from their own perspective is essential to strengthening pandemic responses now and in the future.

To address these evidence gaps, our study sought to document experiences of people with disabilities and their caregivers during COVID-19, in Bangladesh and Liberia. We prioritise the views of people with disabilities as co-researchers within the study as they documented their experiences and directed how they want their stories to be told through photovoice. Photovoice is a participatory method which has its roots in social justice and is often used with marginalised groups with varying literacy levels, from women's health in rural China (Wang and Burris 1997), children in informal settlements in Nairobi (Karuga et al. 2022), to adults with intellectual disabilities (Povee et al. 2014). By drawing on this method, and as guided by feminist epistemologies, we sought to challenge existing power hierarchies that are inherently embedded in western knowledge production, to centre the views and values of people with disabilities. We also draw on feminist epistemologies within our analysis through the application of intersectionality theory, to consider the way in which social factors and structural processes interact with identity-based characteristics to shape individual experiences. For example, we consider how gender, age, disability and socioeconomic status can overlap, to create unique positions of power and privilege leading to both positive and negative experiences during the COVID-19 pandemic (Dean et al. 2017; Tolhurst et al. 2012). Through participatory enquiry, we aim to shift the power balance from researchers and academics holding the authority on knowledge, towards individuals often implicated in systems of oppression, as active participants in research and knowledge production, as experts of their own lives.

## 2. Methods

### 2.1. Approach

This study utilised the participatory research method of photovoice with people with disabilities and caregivers. Photovoice, initially developed by Wang and Burris (1997), originates from community-based participatory research (CBPR) approaches and is a collective visual method whereby members of a community use photographs and storytelling to capture and represent issues important to them in order to initiate community-led action (Budig et al. 2018). We refer to people with physical and psychosocial disabilities, as well as caregivers, as co-researchers because they are participants with lived experiences who

were actively part of the research process, from forming research questions, data collection, data analysis and dissemination. Co-researchers directly shared their photos and stories, which captured their experiences of living through COVID-19.

*2.2. Co-Researcher Selection and Study Sites*

Co-researchers were purposively selected to ensure maximum variation in age, gender and type of disability across the whole sample. Co-researchers were identified through established networks within the National Union of Disabled People's Organisation in Liberia and BRAC University, James P Grant School of Public Health (BRAC JPGSPH) in Bangladesh. Informal caregivers were identified through snowball sampling by working closely with support services. Nine adults with disabilities and four caregivers were selected in Liberia, and seven adults with disabilities and seven caregivers were selected in Bangladesh, with a total of 27 participants in the study (see Table 1). While three informal caregivers were related to selected photovoice co-researchers, as indicated in Table 1, the majority of informal caregivers included in the study were caregivers of relatives with psychosocial disabilities not included as photovoice participants. Eighteen years was chosen as the age of maturity in Bangladesh and Liberia, where participants are legally able to give consent. Study sites were selected with a focus on urban areas, in both Bangladesh and Liberia, as urban areas, such as densely populated main cities and districts, were most affected by COVID-19 during the time of the study. In Liberia, co-researchers were selected from Monrovia, while in Bangladesh, co-researchers were selected from Dhaka, Khulna and Sylhet Divisions through existing links with partners.

**Table 1.** Photovoice Co-researchers.

| Co-Researcher Name | Gender | Type of Impairment/Caregiver | Age | Socioeconomic Status | Location |
|---|---|---|---|---|---|
| **Liberia** | | | | | |
| Heylove R. Mark Sr | Male | Physical impairment | 25–49 years | Middle-income family | Monrovia, Montserrado County |
| Boakai A. Nyehn Jr | Male | Physical impairment | 25–49 years | Middle-income family | Monrovia, Montserrado County |
| Francis C. Sibley | Male | Visual impairment | 25–49 years | Middle-income family | Monrovia, Montserrado County |
| Jochebad Morweh | Female | Physical impairment | 25–49 years | Middle-income family | Monrovia, Montserrado County |
| Benjamin Ballah | Male | Psychosocial impairment | 25–49 years | Middle-income family | Monrovia, Montserrado County |
| Sadiatu Kamara | Female | Physical impairment | 25–49 years | Middle-income family | Monrovia, Montserrado County |
| Rose Dargbe | Female | Physical impairment | 25–49 years | Middle-income family | Monrovia, Montserrado County |
| Patience Duonnah | Female | Hearing impairment | 25–49 years | Middle-income family | Monrovia, Montserrado County |

**Table 1.** *Cont.*

| Co-Researcher Name | Gender | Type of Impairment/Caregiver | Age | Socioeconomic Status | Location |
|---|---|---|---|---|---|
| **Liberia** | | | | | |
| David Hne Wallace | Male | Physical and psychosocial impairment | Over 49 years | Lower-income family | Monrovia, Montserrado County |
| Susan K.C. Nyehn | Female | Caregiver of Boakai, husband with physical impairment | 25–49 years | Middle-income family | Monrovia, Montserrado County |
| Eric Solomon Biawogee | Male | Caregiver of best friend with physical impairment | 25–49 years | Middle-income family | Monrovia, Montserrado County |
| Jonathan Corlon | Male | Caregiver of relative with physical impairment | 25–49 years | Middle-income family | Monrovia, Montserrado County |
| Janet M. Kai | Female | Caregiver of Heylove, partner with physical impairment | 25–49 years | Middle-income family | Monrovia, Montserrado County |
| **Bangladesh** | | | | | |
| Ashraful Alam | Male | Physical impairment | 25–49 years | Middle-income family | Dhaka District, Dhaka Division |
| Marjana Binte Forhad | Female | Physical impairment | 18–24 years | Middle-income family | Faridpur District, Dhaka Division |
| Ataur Islam | Male | Physical impairment | 25–49 years | Middle-income family | Narayanganj District, Dhaka Division |
| Delowar Hossain | Male | Physical impairment | 25–49 years | Lower-income family | Khulna District, Kulna Division |
| Halima Akter | Female | Physical impairment | 18–24 years | Middle-income family | Dhaka District, Dhaka Division |
| Shomoy Chowdhury | Male | Physical impairment | 25–49 years | Higher-income family | Dhaka District, Dhaka Division |
| Saddam Hossian | Male | Physical impairment | 25–49 years | Lower-income family | Manikganj, Dhaka Division |
| Sonia Akter | Female | Caregiver of son with cerebral palsy | 25–49 years | Middle-income family | Dhaka District, Dhaka Division |
| Fatema Rahman Sumi | Female | Caregiver of sister-in-law with psychosocial impairment | 25–49 years | Higher-income family | Dhaka District, Dhaka Division |
| Mehedi Hasan Bokul | Male | Caregiver of brother with hearing and speech impairment | 25–49 years | Middle-income family | Dhaka District, Dhaka Division |
| Bisshojit | Male | Caregiver of sister with psychosocial impairment | 25–49 years | Lower-income family | Khulna District, Kulna Division |
| Nasrin Begum | Female | Caregiver of son with psychosocial impairment | 25–49 years | Middle-income family | Dhaka District, Dhaka Division |
| Bithi Akter | Female | Caregiver of son with cerebral palsy | 25–49 years | Middle-income family | Dhaka District, Dhaka Division |
| Israt Jahan Isha | Female | Caregiver of Marjana, sister with physical impairment | 18–24 years | Middle-income family | Faridpur District, Dhaka Division |

### 3. Data Collection

Due to COVID-19 and lockdowns in place in Bangladesh and Liberia, the photovoice process was adapted to be used remotely, therefore co-researchers were selected who had access to smart phones. All co-researchers were initially provided with training on photovoice to provide an overview of the research, ethics of taking photos and how to use cameras. Training was conducted through videos shared on WhatsApp and in a one-to-one session with researchers, where appropriate and possible to maintain physical distancing. The photovoice steps we conducted are adapted from Ronzi et al. (2019): (1) Co-researchers were asked to take pictures within their surroundings that represent their experience of living through COVID-19 over a three-week period; (2) Co-researchers were asked to share key photos and captions every few days using WhatsApp that focused on the meaning and importance of the photographs. Further clarification and probing was conducted over WhatsApp or phone calls; (3) Co-researchers were asked to pick 10 key photos; (4) Photos were clustered by theme collaboratively between the research team and co-researchers and key photos were selected representing each theme, following validation by co-researchers; (5) Photos were disseminated in a photo booklet.

### 4. Analysis

Photos and captions were received via text and voice notes. Captions were transcribed and translated verbatim by the researchers. The photos and associated captions were validated, organised and summarised according to themes with co-researchers through WhatsApp group conversations and telephone calls. Co-researchers collectively looked at photos and captions, sorting and grouping data representing similar meanings into themes using thematic analysis (Braun and Clarke 2006; Ronzi et al. 2016). We used an inductive framework approach, through sorting data with coding and then explaining links between codes and themes (Ritchie and Spencer 2002); the themes generated are detailed in Table 2. We took an intra-categorical approach to intersectional analysis whereby we considered divergent experiences within 'disabled populations' across Bangladesh and Liberia (Christensen and Jensen 2012). To consider variations in experience within this larger population group, using an intersectional lens, we included a diverse sample of co-researchers with maximum variation to enable analysis into how experiences and inequities are shaped by identity-based characteristics; we used sampling criteria to get a diverse group, by location, age, gender and type of disability, to understand vulnerabilities across groups. Intersectionality places emphasis on understanding lived experience as shaped by the interactions and interconnections of different social markers, such as gender, age, class, ethnicity, sexuality, geography and disability/ability (Green et al. 2017). These interactions occur within structures and systems of power, such as policies or lack of polices. We use intersectionality because it considers inequalities and health disparities as also being produced by underlying power structures, rather than only resulting from risk factors, such as having a disability. We aimed to prioritise the voices of the individual in our intersectional analysis, using a non-additive approach, with emphasis on experiences being unique to co-researchers and their individual and identity-based characteristics (Dean et al. 2017), considering how individual factors shaped the experiences of co-researchers during COVID-19. All photos and themes are presented within Supplementary Material File S1.

**Table 2.** Table of themes generated.

| Themes | Sub-Themes |
|---|---|
| Social connection | • Loneliness<br>• Isolation<br>• Stronger bonds with family |
| Accessibility and awareness | • Health care access and barriers<br>• Mobility challenges<br>• Barriers to health information<br>• Fear of COVID-19 |
| Impact on livelihoods | • Loss of business<br>• Food insecurity<br>• Loss of education<br>• No longer being able to afford medicine<br>• Loss of provider roles |
| Gendered impacts | • Stigma<br>• Marriage<br>• Assumptions |
| Adaptability and technology | • Online school<br>• Access to phones<br>• Online meetings |
| Faith and Nature | • Loss and hope<br>• Places of worship<br>• Green spaces<br>• Environment |

## 5. Results

### 5.1. Impact on Livelihoods

COVID-19 led to economic pressures for many households across both settings for co-researchers with disabilities, as well as caregivers, due to the loss of income from disruptions to work and increased spending. Prior to the pandemic, co-researchers described that, due to their disability, they were more likely to be living in poverty and have lower incomes than people without disabilities, due to health expenses and barriers to livelihood opportunities; these are circumstances which the pandemic has exacerbated. Livelihoods were disrupted significantly. Many co-researchers expressed that they could no longer work; this included businesses being shut, no longer being able to tutor or work on farms. COVID-19 was described as posing an additional barrier to economic and academic advancement for both co-researchers in Bangladesh and Liberia; a metaphor of a closed door to opportunities was used to express obstructions to education and careers. Caregiver co-researchers expressed their distress at no longer being able to support their families; this included mothers, as well as brothers, describing their concerns of being unable to provide care and take on responsibilities for their families. A mother described how she is no longer able to afford medicine for her son with cerebral palsy, while Bisshojit, the older brother of his sister with psychosocial disabilities, expressed his worry about not being able to provide for his family due to his studies being disrupted.

Food insecurity was experienced widely across co-researchers, due to loss of income in already fragile contexts. Many often depended on friends and relatives for support as highlighted in Figure 1. Lower-middle-income families in Bangladesh described experiencing financial crises and feeling unable to share their circumstances with others. Financial hardships also meant that many faced barriers in implementing hygiene measures, as hand sanitisers and buckets were expensive and not accessible nor affordable for all.

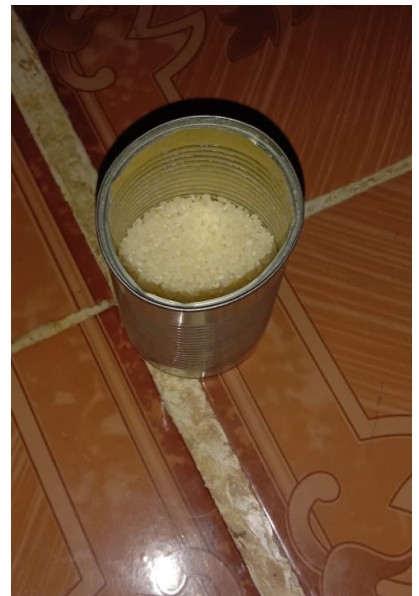

*'One day during the lockdown, we had only a half cup of rice left in the bag for meal. Imagine a disabled person who could navigate when there was free movement. Life becomes very difficult for people with disabilities during natural disasters or pandemics. The fact remains, many persons with disabilities cannot do very hard labour work to earn money that will keep them and their families going. I am a million times grateful to my good neighbour who provided me 10 cups of rice that kept my family and I for a period of time.'*

*Boakai Nyehn, young male, Monrovia, Liberia*

**Figure 1.** Alt-text: A tin half-filled with rice grains.

*'We have been in a financial crisis since last year due to the COVID-19 epidemic. Living in a lower middle income family, we cannot share our problems with others. We have faced trouble to afford my son's regular medicine for his special needs. I joined as a Nursing Officer in Dhaka Medical College last year but still I did not get any salary from there. Now I cannot go to work due to the lockdown.'* (Bithi Akter, 30-year-old female caregiver of son with cerebral palsy, Bangladesh)

*5.2. Gendered Impacts*

Experiences of disability are inherently gendered. For example, perceptions about the origins of disability often led to women experiencing stigma and discrimination, as caregiver co-researchers mentioned they were blamed if their children were disabled. Mothers of sons with cerebral palsy described facing stigma, expressing how they were blamed for their child's condition and often estranged from their husband and in-laws as depicted in Figure 2. Anticipated stigma was illustrated by female co-researchers, who described feeling reluctant to go outside or to social events due to fear of being discriminated or looked at differently within the community; this highlights the gendered impact on female caregivers when they then stay home to support their relatives. These gendered experiences of disability continued through COVID-19; for example, Fatema, a young female caregiver in Bangladesh, expressed her concerns about her sister-in-law who has psychosocial disabilities getting married. Whilst the perception around the importance of marriage was important generally, the need to enhance the financial security of girls with disabilities through marriage was further prioritised as a result of financial insecurities exacerbated during the pandemic. Gender and disability intersect to create further vulnerabilities and raised questions around consent and capacity; fears and concerns were expressed particularly for women and girls with disabilities who may be vulnerable to forced marriages and abuse.

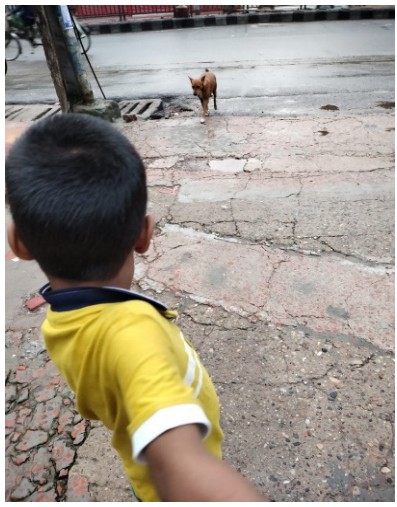

*'Sometimes I feel depressed about my son's uncertain future….Even my in-laws family does not like me. There is ongoing separation between me and my husband after my son's birth. They think that I am responsible for my son's disability.'*

(Bithi Akter, 30-year-old female caregiver of son with cerebral palsy, Bangladesh)

**Figure 2.** Alt-text: A boy in a yellow shirt turning to look at a dog on the road.

*'Munni wants to get married... We feel that she does not have a clear idea about what marriage is . . . I am her elder sister-in-law. I have taken care of her for around 8/9 years. However, we don't want to let her get married to someone. She cannot understand anything properly . . . Is it possible? We do not want to ruin her life.'* (Fatema Rahman Sumi, 32-year-old female caregiver of sister-in-law with psychosocial disability, Bangladesh)

Despite the negative gendered impacts of COVID-19, Marjana explained an incident when she was able to challenge gendered assumptions when she was supporting her father to access care during the pandemic in Figure 3. Marjana described that health professionals at the hospital assumed she was the patient, with one doctor blaming her condition and unmarried status as placing a burden and stress on her father. Her father explained that disabled girls are not burdens to their parents and can also be successful with support.

*5.3. Social Connection*

The juxtaposition of increased family connectedness and loss of social support networks from friends was highlighted by young co-researchers across genders (Figures 4–8). Co-researchers in Bangladesh seemed happy about the additional social connection that COVID-19 brought with family members, particularly in overcoming assumptions about their abilities by showing them skills and attributes which relatives assumed they did not have; for example, being able to cook and help with household chores. Ashraful, as a man, challenged gendered roles by helping in the kitchen, noting that it made him appreciate the hard work that his female relatives do (Figure 7). However, many co-researchers also felt a sense of loss and social connection from peers and friends. Existing services for people with disabilities are often disrupted during crises and this was particularly emphasised by some co-researchers with communication difficulties, as they could no longer see friends and peers they described as being able to understand them, as support group services were cancelled or postponed (Figure 5). Benjamin described how a farming initiative to provide support for people with psychosocial disabilities was also halted during the lockdown in Liberia (Figure 6), while Delowar in Bangladesh was unable to attend his cricket club which provides support for young wheelchair users (Figure 8). Loneliness was felt by many co-researchers, such as Israt and Boakai (Figure 4), leading to negative impacts on mental wellbeing.

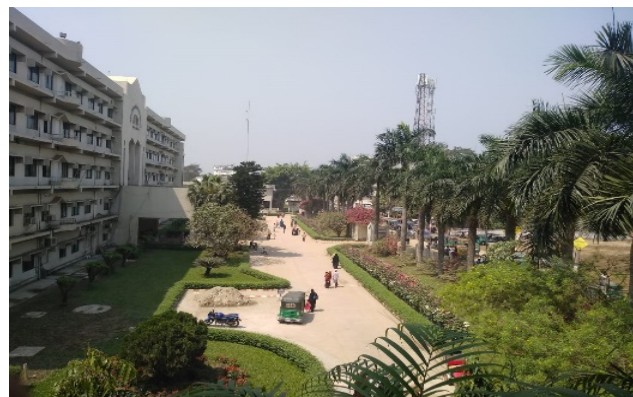

'My father had a mild stroke and I took him to the hospital... Seeing the elbow crutch in my hand everyone thought that I was the patient. No one could have imagined that a disabled girl could take her sick father to hospital. And the interesting thing was the doctor thought that my father had a brain stroke because he was worried about me.

At one point he asked me why I don't get married. My father replied that not all disabled girls are burdens for their father. And if we support the disabled child like other children, they too will be successful in their life. They will also be able to take the responsibility of their parents.'

(Marjana Binte Forhad, 22-year-old female, Faridpur, Bangladesh)

**Figure 3.** Alt-text: A view of a hospital courtyard with a garden, floral bushes and coconut trees.

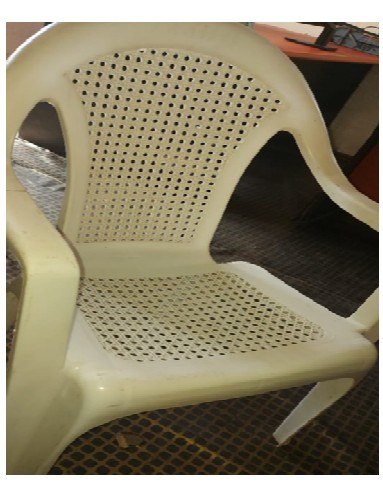

'Experiencing loneliness during this time makes me believe the danger attached to COVID-19 as I remain in the chair sitting in one place for hours during the crisis.'

(Boakai A. Nyehn Jr. younger male, Monrovia, Liberia)

**Figure 4.** Alt-text: An empty white plastic chair.

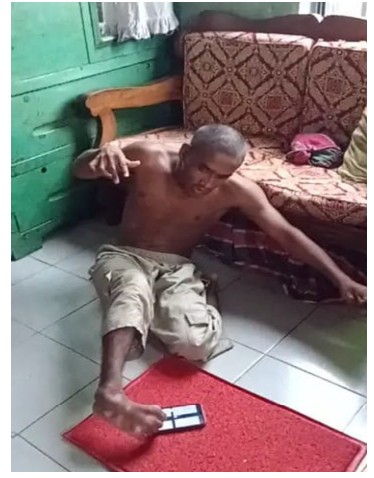

*'You see he uses the mobile phone by using his legs. In this COVID-19 lockdown, he cannot meet with his friends that's why he is trying to call them. He has many friends in the CRP \*, they are also disabled persons. Though he cannot speak properly but his friends understand his language.'*

(Nasrin Begum, 35-year-old female caregiver of son with psychosocial disability) Bangladesh

\* Centre for the Rehabilitation of the Paralysed (CRP) is a non-government organisation of Bangladesh which provides medical treatment, rehabilitation and support services to the persons with disabilities.

**Figure 5.** Alt-text: Young man sitting on a white tiled floor, using his foot to operate his phone.

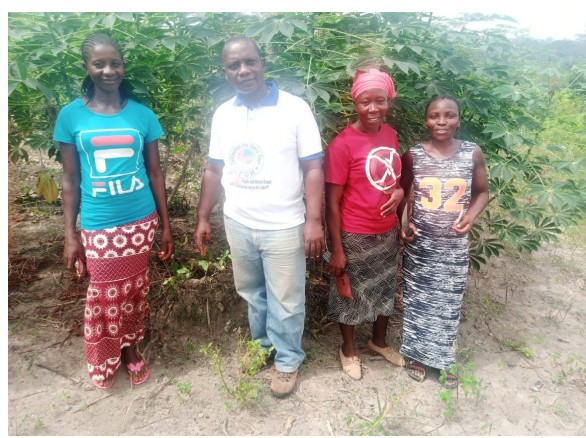

*'This is a photo of people living in recovery from mental health conditions. These people are in Bong County they have grouped themselves to farm as you can see in the photo. This is helpful for their recovery management. Due to the COVID-19 these people were not able to do this'*

(Benjamin Ballah, male, age 40, Liberia)

**Figure 6.** Alt-text: Members of Cultivation for Users Hope, three women and a man, standing together smiling in front of planted trees.

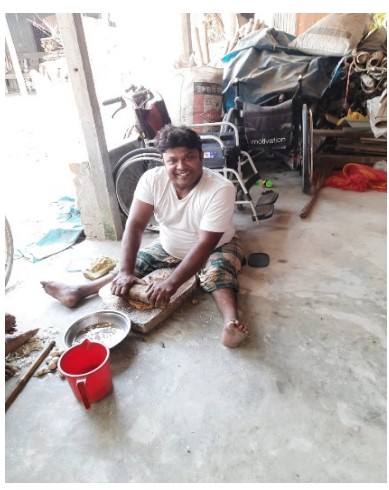

*'I am grinding the spices during this lockdown which is a new experience for me. Before doing this, I thought that I cannot do it but everyone can do anything if they try. It helps me realise the kind of tough jobs wives and mothers regularly do to prepare food for us.'*

(Ashraful Alam, 32-year-old male, Dhaka, Bangladesh)

**Figure 7.** Alt-text: A man sitting on the floor, grinding spices and smiling, with his empty wheelchair behind him.

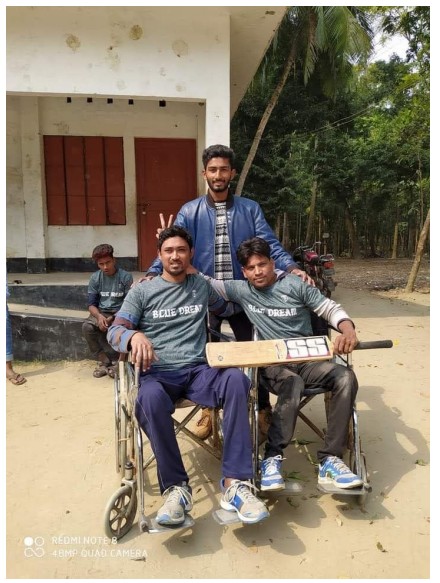

*'I love to play cricket because it helps us to keep our body and mind well. I wanted to develop a cricket organization for the disabled persons in Naral district but it did not happen due to the COVID-19 pandemic'*

(Delowar Hossain, 26-year-old male, Khulna, Bangladesh)

**Figure 8.** Alt-text: Three young men posing outdoors—two young men in wheelchairs smiling and posing with a cricket bat, and another standing behind them.

> *'We are experiencing loneliness during the current coronavirus pandemic. Now we cannot meet with relatives, neighbours or friends to talk about our feelings. Though we are now having a good time with family members, but we are missing our unlocked life. Now we have to rely on technology to communicate with others.'* (Israt Jahan Isha, 18-year-old female, caregiver of sister with physical disability, Bangladesh)

*5.4. Adaptability and Technology*

Technology provided platforms for adaptability in communication, studying and work. However, technology was only empowering for those who had access to smart phones, particularly younger generations of higher socio-economic status who lived in cities. The wide use of technology often excluded people, particularly women living in rural areas, and older population groups who do not have access to smart phones.. Many younger co-researchers living in the cities in Liberia and Bangladesh illustrated how online platforms, such as Zoom, helped them to adapt through online learning and working from home (Figures 9 and 10). However, technology was also described as a barrier, as some co-researchers described experiencing difficulties in online learning, and missing face-to-face contact with teachers.

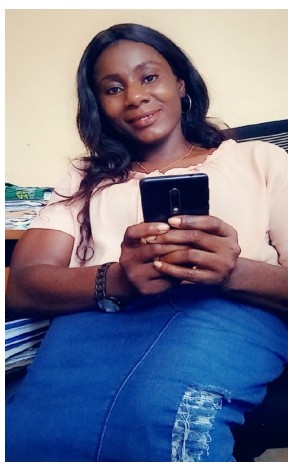

*'The lockdown has made a very good use of technology since people were observing social distancing and working from home'*

(Rose Dargbe, younger female, Liberia)

**Figure 9.** Alt-text: A woman with long hair smiling holding a smartphone.

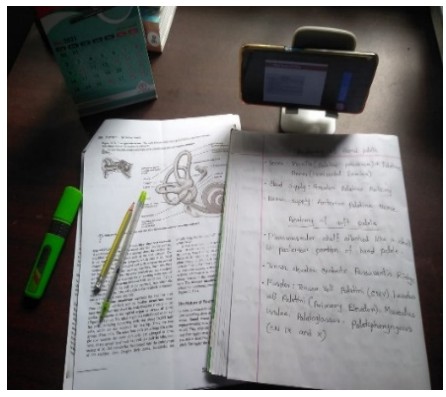

'Online classes started in April last year. After looking at the mobile or laptop screen for one and a half to two hours, can students keep full concentration on the online classes? I cannot do it. In the offline classes, we attended the class in person in front of the teachers. By looking student's faces, the teachers could also see whether students have understood the work but in online classes, this is not possible.'

(Marjana Binte Forhad, 22-year-old female, Faridpur, Bangladesh)

**Figure 10.** Alt-text: A desk with notes, pens and pencils and an online class being streamed on a phone.

*5.5. Accessibility and Awareness*

Accessibility to and comprehension of COVID-19 messaging, either due to hearing impairments or cognitive capacity to act, were barriers across co-researchers. Inaccessibility was mentioned in terms of both communication and mobility. Co-researchers in Liberia mentioned how masks can often isolate the deaf community, as it is difficult to lip read or see facial expressions as mentioned by Rose in Figure 11. Barriers to accessing public health information was also highlighted, as many health messages were broadcast on radios and tv without any interpreters. Fatema, a young female caregiver in Bangladesh, expressed her fears, as her sister-in-law with learning disabilities refused to wear a mask, as she did not understand COVID-19 precautions.

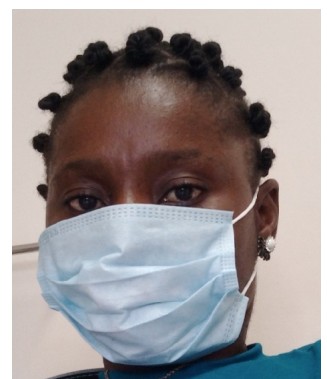

'During the lockdown, the wearing of masks makes it difficult for people who are deaf to get needed information or communicate freely. It is very difficult for a person with hearing impairment to be able to read someone's lips or know what their expressions are when using masks'

(Rose Dargbe, younger female, Liberia)

**Figure 11.** Alt-text: Young woman wearing a blue face mask.

Ongoing mobility challenges were captured across Bangladesh and Liberia; wheelchair users are unable to access roads during the rainy seasons, or access banks. Many co-researchers highlighted how inaccessibility creates further distance and disparities between communities. The intersection of access, geography and socio-economic status was also highlighted. Shomoy, living in Dhaka city, described how a wheelchair ramp in front of his house, with the help of his parents, has enabled him to move independently (Figure 12), while Saddam, living in a peri-urban area, described the lack of disabled-friendly washrooms in his home and therefore having to rely on his mother for bathing (Figure 13).

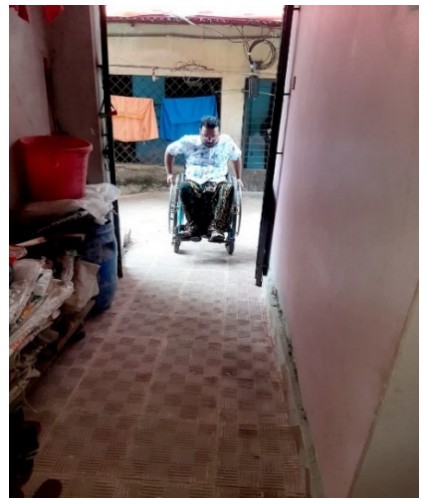

*'There was no wheelchair ramp in front of our house while we rented the house. But my parents requested to our house owner to develop a wheelchair ramp for me. Now I can move around without anyone's help.'*

(Shomoy Chowdhury Shobuj, 31-year-old male, Dhaka, Bangladesh)

**Figure 12.** Alt-text: A man in a wheelchair being able to move freely in the entrance of his home.

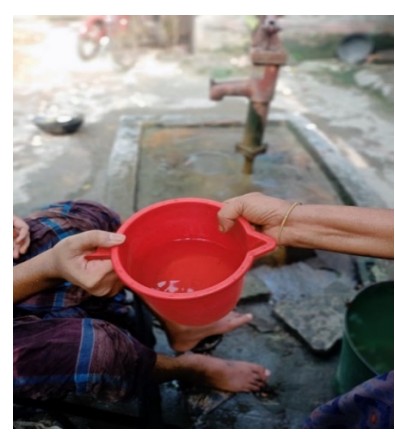

*'As I have no disabled friendly washroom in my house, I always take the support from my mother for bathing. I live in a poor income family and you know that there are no special opportunities for us in village. So this is a daily scenario for me.'*

(Saddam Hossian, 25-year-old male, Manikganj, Bangladesh)

**Figure 13.** Alt-text: A red jug of water being passed from a mother's hand to her son's hand.

While generalised barriers to access remained relating to accessibility, distance and affordability, these were exacerbated during COVID-19 due to the increased pressure on the health care system, as well as a sense of general fear. Barriers to accessing healthcare emerged across Liberia and Bangladesh. Fear of contracting COVID-19 in hospitals or travelling on public transport resulted in many participants not attending health facilities or services. Lockdown and financial difficulties also prevented participants from accessing treatment on time; Delowar, in Bangladesh, explained having to resort to treatment from a government hospital and expressed his concerns of contracting COVID-19 while admitted (Figure 14). Co-researchers in Liberia mentioned how lockdown created challenges for people with disabilities in accessing appropriate treatment and assistive equipment, as access to these services were not prioritised.

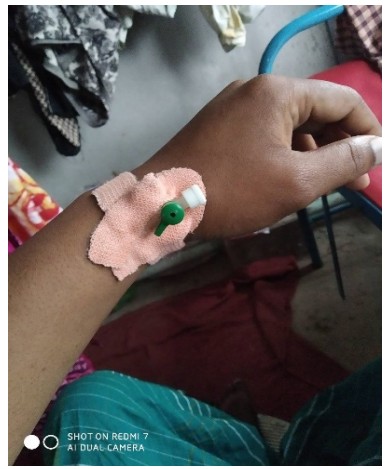

*'I had a brain stroke during the lockdown for coronavirus disease. I did not get proper treatment after my stroke during the initial phase of COVID-19. I passed a very bad time then. I could not go to the hospital immediately due to the lockdown and also due to my financial crisis. Due to my financial instability, I received the treatment from a government hospital. But there was no practice of hygiene maintenance to avoid corona in the hospital…'*

(Delowar Hossain, 26-year-old Male, Narayanganj, Bangladesh)

**Figure 14.** Alt-text: An arm with a cannula bandage in a hospital.

### 5.6. Faith and Nature

Faith was often mentioned as a source of strength and peace across co-researchers in Bangladesh and Liberia. Co-researchers expressed their sadness at not being able to attend places of worship, as they formed communities of care, while some participants also relied on their faith to protect them from COVID-19 (Figures 15 and 16). Men in Bangladesh described feeling emboldened and strengthened by their faith to attend congregational prayers, despite lockdown restrictions more than women, who stayed home to pray. Coresearchers in Liberia and Bangladesh described interaction with nature as positively impacting mental wellbeing. Nature, gardening and green spaces were portrayed as providing safe spaces for reflection, hope and peace of mind (Figure 17). Fruit was also captured by participants, who emphasized the importance of nutrition for strengthening immune systems to protect them from COVID-19. Plants and flowers were often used as metaphors for connection and growth and being able to thrive in the right environment, with adequate support. For example, Rose in Liberia used the watering of flowers as an analogy for encouragement and support to be reconnected to society (Figure 18), while Delowar and Ataur (Figure 19) used plants as a metaphor for disabled people being able to flourish and be established, if they are rooted in supportive environments.

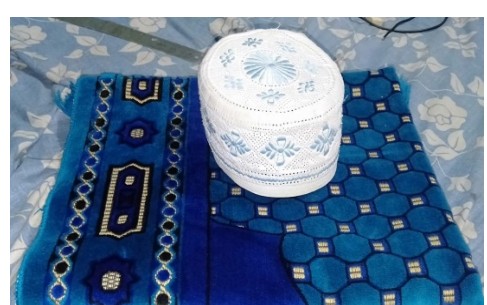

*'Despite my willingness, I have not been able to perform my daily five times prayers in Mosque since last one and half years due to the COVID-19 situation. Even I cannot participate in Salat al-Jumu'ah as instead of the Zuhr prayer on Friday afternoons. That's why, I am trying to say my prayers at home.'*

Ataur Islam, 40-year-old male, Narayanganj, Bangladesh

**Figure 15.** Alt-text: A white and blue embroidered prayer cap on a blue patterned prayer mat.

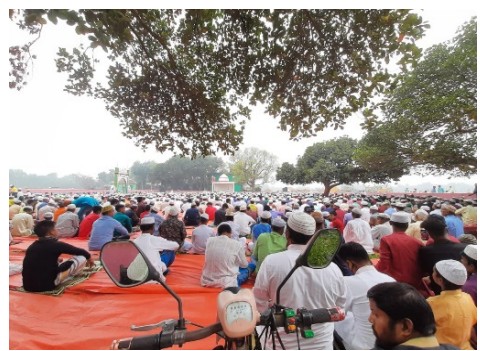

*'Relying on Allah and without caring about COVID-19, I went to the Eid gathering for saying my Eid prayers with everyone. We all prayed to Allah so that we can save ourselves from COVID-19. This was a different experience for me.'*

Ashraful Alam, 32-year-old male, Dhaka, Bangladesh

**Figure 16.** Alt-text: A large crowd of people sitting outdoors for Eid prayers.

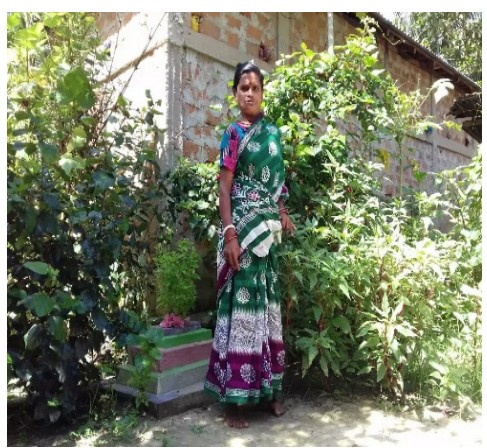

*'A tulsi tree is planted in front of our household. We believe that tulsi plant has sacred power. The tulsi leaves also can reduce our illness like as cold, cough and asthma related diseases by its healing power so my mother and sister try to worship it every day.'*

Bisshojit, 22-year-old male caregiver of sister with psychosocial disability, Bangladesh.

**Figure 17.** Alt-text: A woman dressed in a colourful green, white and purple sari standing in front of bushes and tulsi plants.

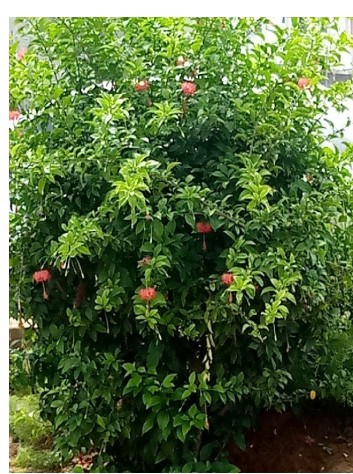

*'This picture of these fresh flowers reminds me of connection and growth. Whatever life's situation is, once we are being watered by encouragement, we will still get reconnected'*

Rose Dargbe, younger female, Liberia

**Figure 18.** Alt-text: A lush green tree with bright red flowers.

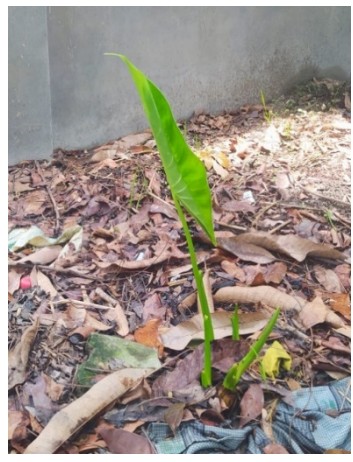

*'As it is our duty to keep the base of the tree clean and give it a chance to grow. Similarly, it is the responsibility of our society and family to support the disabled persons so that they can lead a better and healthier life.'*

Delowar Hossain, 26-year-old male, Khulna, Bangladesh

**Figure 19.** Alt-text: A bright green plant growing in the dirt.

## 6. Discussion

COVID-19 has placed a magnifying glass on inequity and vulnerabilities for disabled people, exacerbating existing barriers in access to services, livelihoods, increased risks, stigma and isolation (Banks et al. 2017; Kuper et al. 2021). Our research, through an intersectional analysis, has highlighted that people with disabilities are not a homogenous group; different types of impairment, gender, cultural identity and socio-economic status interplay to produce different challenges (Goethals et al. 2015). Intersectionality enables deeper understandings of the nuances of inequity and the interconnections between them in different contexts, while also considering wider societal and power structures that shape them (McCollum et al. 2019).

Across two country contexts, there were common barriers faced, including lack of access to COVID-19 information, but this differed by type of impairment. Many experienced isolation and barriers in receiving social protection measures and this was also shaped by different factors; for example, those who lived in more peri-urban areas had less access to employment and subsistence support. Photovoice highlighted challenges and informed ways to develop more inclusive responses using multisectoral action. Communication barriers, particularly in shaping understandings of health information, was highlighted and this has been found in other settings, such as Ghana, where deaf people were not included in any health messaging regarding COVID-19 (Swanwick et al. 2020). To address this, policy makers and service providers should prioritise accessible health information, including interpreters and sign language. This needs to be planned early and include persons with hearing and speech impairments, as well as psychosocial disabilities, including intellectual disabilities, to co-create health information that is accessible for diverse groups using easy-to-read language and images, (Books Beyond Words 2020; Terras and Jarrett 2021). The nexus between disability and poverty was exacerbated for many in both settings and this has also been reflected in studies looking at how intersections of gender, poverty and disability influence the experiences of women and men with disabilities living in low– middle-income settings, particularly in relation to disrupted livelihoods in the informal sector and food insecurity (Banks et al. 2021; Hasan et al. 2021). Rashid et al. (2020) and Banks et al. (2021) framed recommendations using an intersectional framework to broaden social protection packages and support resilience, rather than rely on a deficit model; it is key that the design of assistance and stimulus packages are inclusive of people with disabilities to ensure that they are accessible and relevant.

Social connection and support are important, as highlighted by co-researchers, and the negative psychosocial impact of lockdown has been widely documented (Lund et al. 2020). Therefore COVID-19 responses should consider how networks and support can be maintained in times of health system shocks, and how this may vary for participants based on their identity or type of disability. For example, technology was shown to be a

lifeline for many younger people with disabilities who had access to smartphones; online support services were essential, as demonstrated in a range of studies looking at health in marginalised or remote settings, where online technologies have been effective and cost-effective tools in supporting engagement and communication (Goggin and Ellis 2021; Liegghio and Caragata 2020; López et al. 2018; Zaagsma et al. 2020). However, for many without access to smart phones, adapting support services in outdoor settings with social distancing protocols in place could serve to maintain connection and communication while addressing social isolation. The role of informal caregivers is often overlooked in disability research; however, a key finding in our study was the inclusion of caregivers and their perspectives, including the additional fears, responsibility and stigma often experienced particularly during COVID-19. This further highlights the need for psychosocial support services to also consider caregivers and build networks of support (Bertuzzi et al. 2021).

Understanding the impact of COVID-19 on people with disabilities from their own perspective is essential to strengthening pandemic responses and accountability. Meaningful collaboration with people with disabilities and disabled people's organisations, and the inclusion of people with disabilities in positions of leadership at policy level, is critical, if the needs and priorities of people with disabilities are to be represented. Photovoice enabled an inclusive platform and space to recognise lived experience and knowledge, and the chance to be heard collectively and individually, particularly as disabled people are often neglected from meaningful participation in research and decision-making processes (Liegghio and Caragata 2020). Policy responses during the pandemic must have a disability lens applied to them, with a rights-based approach to highlight the needs of people with disabilities, to develop strategies that ensure no one is left behind (United Nations 2019) in immediate and long-term responses to COVID-19.

## 7. Conclusions

By prioritising the views of people with disabilities and a focus on co-production, we aimed to challenge existing power hierarchies that are inherently embedded in western knowledge production as co-researchers directed how they wanted their stories to be told,. People with disabilities should be placed at the centre of the response, participating as agents of planning and implementing policies, shaped by their priorities. Photovoice was a powerful tool in placing the voices of disabled people at the forefront and adding to an evidence base to inform inclusive responses to pandemics. A strength of our study was being able to adapt our methods innovatively to use remote technologies, despite the pandemic and lockdowns. WhatsApp groups enabled a virtual space to build connections and discussions and photos were shared collectively. However, this was also a limitation, as this excluded people who did not have access to smartphones and those living in rural areas.

Our research underlines how accessibility is fundamental to the inclusion of people with disabilities in the health and socio-economic response to COVID-19. Ensuring accessibility of information and services is key, as well as addressing structural and social barriers to enable people with disabilities to have equitable access to services. This paper presents lived experiences through an intersectoral lens, highlighting stories of people with disabilities and caregivers, who are often seldom heard. Understanding areas that need to be prioritised and considered from the perspectives of people with disabilities can inform more equitable responses to health system shocks. We also emphasise the importance of how creative participatory methods such as Photovoice can promote meaningful engagement and participation of marginalised groups in research to foster advocacy and social change.

**Supplementary Materials:** The following supporting information can be downloaded at: https://www.mdpi.com/article/10.3390/socsci11090378/s1, Supplementary Material File S1: Pandemic Portraits Photo Booklet.

**Author Contributions:** Conceptualization, S.C.; methodology, S.C., S.A.U., B.A.N.J., H.R.M.S., L.D., N.B.H.; validation, S.C., S.A.U., B.A.N.J., H.R.M.S., L.D; formal analysis, S.C., S.A.U., B.A.N.J., H.R.M.S., L.D.; writing—original draft preparation, S.C., S.A.U., B.A.N.J., H.R.M.S., L.D.; writing— review and editing; S.C., S.A.U., B.A.N.J., H.R.M.S., M.T.H., S.F.R., N.B.H., L.D.; supervision, M.T.H., S.F.R., N.B.H., L.D. All authors have read and agreed to the published version of the manuscript.

**Funding:** This research was funded by the Royal Society of Tropical Medicine and Hygiene (RSTMH) Small Grants Program 2020 (NZ6 AGB). This was a collaborative project with the Liverpool School of Tropical Medicine (LSTM), the National Union of Disabled People's Organisations (NUOD), Liberia, and BRAC University, James P Grant School of Public Health (BRAC JPGSPH), Bangladesh. Some funding from the following programmes has supported staff salaries on the grant to produce the publication: the NIHR-funded NIHR RIGHT-Funded Research Consortium REDRESS: Reducing the Burden of Severe Stigmatising Skin Diseases in Liberia, grant reference: NIHR200129 and the UKRI GCRF-funded Accountability for Informal Urban Equity Hub (ARISE), which is a UKRI Collective Fund award, RC grant reference: ES/S00811X/1.

**Institutional Review Board Statement:** Ethical approval was granted from the Liverpool School of Tropical Medicine (20-040), the University of Liberia, the Pacific Institute for Research and Evaluation Institutional Review Board (UL-PIRE IRB) (No. 22-03-310 with assurance No. FWA00032198) and by the Institutional Review Board (IRB) of BRAC James P Grant School of Public Health, BRAC University Bangladesh (No: 2019-034-IR).

**Informed Consent Statement:** Informed consent was obtained from all subjects involved in the study and written informed consent has been obtained from all co-researchers to publish this paper, including their names and images. Prior to selection, the study was explained to all participants verbally over the phone, as well as with an information sheet which was sent via WhatsApp for those who were literate. All participants were given the opportunity to ask questions and reassured that they could withdraw from the study at any stage. Considering the vulnerability of co-researchers, safeguarding was in place; for example, if in the case individuals required further support, resulting from issues raised through the data collection processes, care would be taken to match participants with appropriate support services, offered through local organisations and existing health system links. We also ensured that the research team had training regarding safeguarding, disability inclusion and adaptation of communication methods.

**Data Availability Statement:** Not applicable.

**Acknowledgments:** We would like to extend our deepest gratitude to our co-researchers for their time and trust in openly and creatively sharing their personal experiences and stories: Ashraful Alam, Marjana Binte Forhad, Ataur Islam, Delowar Hossain, Halima Akter, Shomoy Chowdhury, Sonia Akter, Fatema Rahman Sumi, Mehedi Hasan Bokul, Bisshojit, Nasrin Begum, Bithi Akter, Israt Jahan Isha, Saddam Hossian, Benjamin Ballah, Sadiatu Kamara, Rose Dargbe, Francis C. Sibley David Hne Wallace, Patience Duonnah, Jochebad Morweh, Susan K. C. Nyehn, Eric Solomon Biawogee, Jonathan Corlon, and Janet M. Kai. Our many thanks to the late Naomi B Harris; we dedicate this work to you for your dedication and courageous leadership in advocating for disability rights in Liberia and beyond—thank you for your mentorship and guidance.

**Conflicts of Interest:** The authors declare no conflict of interest. The funders had no role in the design of the study; in the collection, analyses, or interpretation of data; in the writing of the manuscript, or in the decision to publish the results.

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
