# Peer review of "Pandemic Portraits—An Intersectional Analysis of the Experiences of People with Disabilities and Caregivers during COVID-19 in Bangladesh and Liberia"

_socsci, doi:10.3390/socsci11090378_

Round 1

Reviewer 1 Report

Overall, I am impressed by this research demonstrating the difficulties that disabled people faced during the COVID-19 pandemic. The data used in this study are abundant and thoughtful. Here I provide some suggestions to improve the article.

Though the intersectionality theory is often applied to the study of social equity, it would be great if more information can be provided to support how the theory is applied to the current study, and how the analyses are guided by the theory. Also, it would be better if the authors can provide further information (in tables perhaps) to reveal the processes of thematic analysis.

Relatedly, the intersectionality of the experience seems vague for certain themes. For example, regarding social connection, did all coresearchers experience both increased family connectedness and reduced social support from friends? If not, who were likely to experience both, or only one, or the other? Similar questions arise for themes of technology and faith. Who were likely empowered by the technology and who experienced barriers of using it? Who tended to develop faith and hope and who did not?

Regarding table 1, I am wondering if it is ethical, or necessary, to show up names of the participants in a research article. Personal identification of participants is not allowed in most research. 

Author Response

Dear Reviewer,

We would like to express our gratitude for your valuable suggestions and feedback. Thank you for your time and thorough review, your comments have strengthened the presentation and content of our paper. Please kindly find edits attached in the document and responses below in red.

Overall, I am impressed by this research demonstrating the difficulties that disabled people faced during the COVID-19 pandemic. The data used in this study are abundant and thoughtful. Here I provide some suggestions to improve the article.

Though the intersectionality theory is often applied to the study of social equity, it would be great if more information can be provided to support how the theory is applied to the current study, and how the analyses are guided by the theory. Also, it would be better if the authors can provide further information (in tables perhaps) to reveal the processes of thematic analysis.

Thank you so much for your valuable suggestions and feedback. This has been included in Table 2 and the application of intersectionality theory has been expanded upon within the text (lines 75-88, and further in the analysis section 144-173).

Relatedly, the intersectionality of the experience seems vague for certain themes. For example, regarding social connection, did all coresearchers experience both increased family connectedness and reduced social support from friends? If not, who were likely to experience both, or only one, or the other? Similar questions arise for themes of technology and faith. Who were likely empowered by the technology and who experienced barriers of using it? Who tended to develop faith and hope and who did not?

Many thanks, this has been drawn out more explicitly within the themes in the Results section (lines 277-300, 318-323, 383-385).

Regarding table 1, I am wondering if it is ethical, or necessary, to show up names of the participants in a research article. Personal identification of participants is not allowed in most research.

We included names of co-researchers as we obtained consent to publish names in publications and photo booklets. Participants stated that they wanted to be named and credited for their photos, and this has been validated by all co-researchers.

Reviewer 2 Report

Dear authors:

Congratulations for the work presented, for the methodology used and for the intended purpose. It is a study clearly focused on improving the living conditions of t oppressed people and producing changes in power relations.

The supplementary material provided should also be valued.

Only some considerations are indicated in case they can contribute to the improvement of the study:

Introduction section: it is recommended to specify how feminist epistemology has been included in the study.

It is recommended to include more support on other research developed with photovoice with similar objectives. 

It is recommended to indicate a definition of "urban spaces" adapted to the working context.

Method section: in Table 1 it is recommended to include age and socioeconomic status. It is recommended to specify if the names of the co-researchers are pseudonyms.

It is recommended to specify whether the participating co-investigators, caregivers and persons with disabilities, are related to each other in the sense of whether they are the caregivers of the participating persons with disabilities. It is recommended to explicitly state that they are informal caregivers and to include some reference to the implications of being an informal caregiver.

Analysis section: it is recommended to include information on sampling ("diverse sample of coresearchers with maximum variation to enable...") in section 2.2.

When mentioning that a "thematic analysis" is performed, it is recommended that reference be made to the type of thematic analysis being performed.

It is recommended to specify how the intersectional analysis is carried out at the methodological level and to include references.

Results section: it is recommended to include the system of categories generated from the analysis of the information, specifying the themes and sub-themes. If possible, it is recommended to specify whether the information emerges from the caregivers or from the persons with disabilities.

Conclusions section: it is recommended to include in a clearer way information on weaknesses and strengths of the study. Likewise, it is suggested to include future perspectives beyond the fact that the study allows identifying new ways of responding to pandemics.

References within the text: a revision is needed so that they appear in alphabetical order within the parentheses.

Author Response

Dear Reviewer,

We would like to express our gratitude for your valuable suggestions and feedback. Thank you for your time and thorough review, your comments have improved the presentation and content of our paper. Please kindly find edits attached in the document and responses below in red.

Congratulations for the work presented, for the methodology used and for the intended purpose. It is a study clearly focused on improving the living conditions of t oppressed people and producing changes in power relations.

The supplementary material provided should also be valued.

Only some considerations are indicated in case they can contribute to the improvement of the study:

Introduction section: it is recommended to specify how feminist epistemology has been included in the study.

It is recommended to include more support on other research developed with photovoice with similar objectives. 

It is recommended to indicate a definition of "urban spaces" adapted to the working context.

Thank you so much for these valuable suggestions, how feminist epistemology has been included in the study and more examples of research developed with photovoice with a focus on marginalised groups has been included in the introduction, as attached (lines 71-88), with expansion on urban spaces in the methodology (lines 116-117).

Method section: in Table 1 it is recommended to include age and socioeconomic status. It is recommended to specify if the names of the co-researchers are pseudonyms.

It is recommended to specify whether the participating co-investigators, caregivers and persons with disabilities, are related to each other in the sense of whether they are the caregivers of the participating persons with disabilities. It is recommended to explicitly state that they are informal caregivers and to include some reference to the implications of being an informal caregiver.

We included names of co-researchers as we obtained consent to publish names in publications and photo booklets. Co-researchers stated that they wanted to be named and to be credited for their photos, and this has been validated by all co-researchers. Age and socio-economic status have been included as well as caregiver roles explicitly stated in table one, as well as within lines 111-113.

Analysis section: it is recommended to include information on sampling ("diverse sample of coresearchers with maximum variation to enable...") in section 2.2.

When mentioning that a "thematic analysis" is performed, it is recommended that reference be made to the type of thematic analysis being performed.

It is recommended to specify how the intersectional analysis is carried out at the methodological level and to include references.

Many thanks for this. Expansion on sampling, thematic analysis and intersectional analysis has been included in the analysis (lines 144-173).

Results section: it is recommended to include the system of categories generated from the analysis of the information, specifying the themes and sub-themes. If possible, it is recommended to specify whether the information emerges from the caregivers or from the persons with disabilities.

Thank you for this suggestion. This has been included in Table 2 and expanded upon within the Results section.

Conclusions section: it is recommended to include in a clearer way information on weaknesses and strengths of the study. Likewise, it is suggested to include future perspectives beyond the fact that the study allows identifying new ways of responding to pandemics.

Limitations of this study and further implications of the study have now been included in the conclusion (lines 480-485).

References within the text: a revision is needed so that they appear in alphabetical order within the parentheses. 

This has been revised accordingly within the text.
